computational chemistry/photochemistry/physical chemistry

photolysis, peroxide radicals, complete active space 2nd order perturbation theory (CASPT2), *ab initio*

**Authors for correspondence:**
Rashid R. Valiev
e-mail:valievrashid@gmail.com
Theo Kurten
e-mail: theo.kurten@helsinki.fi

This article has been edited by the Royal Society of Chemistry, including the commissioning, peer review process and editorial aspects up to the point of acceptance.

# Is either direct photolysis or photocatalysed H-shift of peroxyl radicals a competitive pathway in the troposphere?

## Rashid R. Valiev[1,2] and Theo Kurten[1,3]

[1]Department of Chemistry, University of Helsinki, PO Box 55 (A.I. Virtanens Plats 1), 00014 Helsinki, Finland
[2]Tomsk State University, 36, Lenin Avenue, 634050 Tomsk, Russia
[3]Institute for Atmospheric and Earth System Research, University of Helsinki, Helsinki 00014, Finland

RRV, 0000-0002-2088-2608

Peroxyl radicals ($RO\dot{O}$) are key intermediates in atmospheric chemistry, with relatively long lifetimes compared to most other radical species. In this study, we use multireference quantum chemical methods to investigate whether photolysis can compete with well-established $RO\dot{O}$ sink reactions. We assume that the photolysis channel is always $RO\dot{O} + h\nu => RO + O(^3P)$. Our results show that the maximal value of the cross-section for this channel is $\sigma = 1.3 \times 10^{-18}\,cm^2$ at 240 nm for five atmospherically representative peroxyl radicals: $CH_3O\dot{O}$, $C(O)HCH_2O\dot{O}$, $CH_3CH_2O\dot{O}$, $HC(O)O\dot{O}$ and $CH_3C(O)O\dot{O}$. These values agree with experiments to within a factor of 2. The rate constant of photolysis in the troposphere is around $10^{-5}\,s^{-1}$ for all five $RO\dot{O}$. As the lifetime of peroxyl radicals in the troposphere is typically less than 100 s, photolysis is thus not a competitive process. Furthermore, we investigate whether or not electronic excitation to the first excited state ($D_1$) by infrared radiation can facilitate various H-shift reactions, leading, for example, in the case of $CH_3O\dot{O}$ to formation of $\dot{O}H$ and $CH_2O$ or $HO\dot{O}$ and $CH_2$ products. While the activation barriers for H-shifts in the $D_1$ state may be lower than in the ground state ($D_0$), we find that H-shifts are unlikely to be competitive with decay back to the $D_0$ state through internal conversion, as this has a rate of the order of $10^{13}\,s^{-1}$ for all studied systems.

# 1. Introduction

Photolysis processes play an important role in the Earth's atmosphere, because they lead to the formation of radicals, ions, atoms and other reactive species [1]. In the troposphere,

the photodissociation of $O_3$, HONO and $H_2O_2$ leads to the formation of OH radicals [2], while photolysis of $NO_2$ leads to the formation of $O(^3P)$ (and thus $O_3$) [3]. The photolysis of $O_2$ is also the main mechanism creating the ozone layer in the stratosphere [4]. Photolysis is an important loss process for a few organic species in the troposphere, such as formaldehyde [4], or acetone and other small ketones [5]. However, for most closed-shell organic compounds, the role of photolysis is more indirect: rather than being directly photolysed, the dominant atmospheric loss process, for example, for hydrocarbons is a bimolecular reaction with one of the photochemically generated oxidants (e.g. $\dot{O}H$, $O_3$ or $NO_3$). With the exception of $NO_2$, the photolysis of radical species is seldom considered in atmospheric chemistry, mainly due to the very short lifetimes of most atmospherically relevant classes of radicals. For example, most alkoxy ($R\dot{O}$) and alkyl ($\dot{R}$) radicals typically have atmospheric lifetimes around or below $10^{-4}$ and $10^{-8}$ s, respectively [6–8]. Even if photolysis of these species were reasonably fast, it would thus be unable to compete with other chemical loss channels [1,5,6].

Peroxyl radicals ($RO\dot{O}$), usually formed by the addition of $O_2$ to carbon-centred radicals, are crucial intermediates in both atmospheric oxidation and combustion. Compared to alkoxy and alkyl radicals, peroxyl radicals are relatively long-lived, as their main sink reactions involve low-concentration radicals such as NO, $HO\dot{O}$ or other $RO\dot{O}$ [7–10]. Especially, the latter reaction mechanism has recently received much attention as it may lead (via intersystem crossings) to the formation of low-volatility ROOR 'dimers' crucial for the growth of atmospheric aerosol [10]. Complex peroxyl radicals have also recently been found to have unimolecular reaction channels, such as H-shifts, but these are seldom much faster than $1 \, s^{-1}$ at atmospheric temperatures [11]. In general, the tropospheric lifetimes of $RO\dot{O}$ are thought to vary between about 0.01 and 100 s [11], with longer lifetimes corresponding to cleaner (unpolluted) conditions. If photolysis of $RO\dot{O}$ were exceptionally rapid, e.g. comparable to that of $NO_2$, it could thus contribute substantially to $RO\dot{O}$ loss in clean conditions. Photolysis of $RO\dot{O}$ has been studied both in the UV and the IR spectral regions [12–16]. Experimental cross-sections for photolysis in the UV region have been presented for multiple peroxyl radicals [15,17–28] and compiled, for example, in the dataset 'MPI-Mainz UV/VIS spectral atlas of gaseous molecules of atmospheric interest' [29]. However, the actual rate constant of $RO\dot{O}$ photolysis has, to our knowledge, not been directly estimated by theoretical or experimental methods. In addition to potential UV photolysis, Frost [12] has proposed that photolysis of $CH_3O\dot{O}$ (or other $RO\dot{O}$) may also occur indirectly via IR excitation to the first excited state ($D_1$). Even though the $D_1$ state is bound, Frost [12] proposed that for example the 1,3 H-shift of the $CH_3O\dot{O}$ radical in this state is extremely fast, leading to rapid formation of OH and $H_2CO$. In the ground state, the 1,3 H-shift is prevented by a prohibitively high energy barrier. However, Frost did not explicitly compute the lifetime for the $D_1$ state, but simply assumed it to be long enough to allow for the reaction. We note that for larger $RO\dot{O}$, multiple different H-shift channels may be available. Some of these lead to prompt dissociation (as in the case of $CH_3O\dot{O}$), while others lead to alkyl radicals (denoted 'QOOH' in combustion chemistry), which may then add $O_2$ to yield second-generation peroxyl radicals [11]. In both cases, enhancement of the H-shift rates by IR excitation would be of great relevance to atmospheric chemistry.

In order to calculate the rate constant of photolysis, the solar flux and the wavelength-dependent photolysis cross-section are required [1,30,31]. The cross-section can be obtained either from measurements or by quantum chemical calculations [31,32]. The calculation of photolysis cross-sections of polyatomic molecules is computationally demanding [31,33]. Recently, a simple and fast model was developed for calculating photolysis cross-sections [34]. This model was applied to many diatomic molecules, including oxides, fluorides and chlorides of alkali metals [34–36]. The photolysis rate constants for these species were calculated in the exospheres of the Moon, Mercury and satellites of Jupiter using the experimentally measured solar flux. In the Earth's troposphere, the solar flux function at wavelengths shorter than 305 nm is cut off by the ozone layer [1,37]. (We note that some radiation in the 290–305 nm interval does reach the troposphere, and is crucial for tropospheric $O_3$ photolysis and $\dot{O}H$ production, but for the purpose of computing $RO\dot{O}$ photolysis rates at ground-level conditions, we ignore this 'tail' as the flux is relatively low.)

The solar flux was measured for typical tropospheric conditions, for example, in the work of Chance & Kurucz [38]. Taking these measurements into account, we have calculated the tropospheric photolysis cross-sections and representative rates for five peroxyl radicals, which contain an atmospherically representative selection of other functional groups (alkyl, carbonyl and acyl). Also, we simulated various H-shift reactions of $RO_2$ in the $D_1$ excited state, and calculated the lifetime of the $D_1$ state with respect to internal conversion (IC; i.e. decay back to the ground state).

# 2. Theoretical model and calculation details

## 2.1. Theoretical model of photolysis

The rate of photolysis $J$ of a molecule by solar photons can be expressed as

$$J = \int_0^{\lambda_{\mathrm{bind}}} \sigma(\lambda)\Phi(\lambda)\mathrm{d}\lambda, \tag{2.1}$$

where $\sigma$ is the photolysis cross-section of the molecule, $\Phi$ is the solar photon flux and $\lambda_{\mathrm{bind}}$ is the wavelength of photons corresponding to the binding energy of the bond broken in the photolysis reaction [30,31]. By applying equation (2.1), we implicitly assume that the quantum yield of photolysis is 1. For the UV excitations studied here, this is a reasonable assumption, as the relevant excited electronic states are unbound.

For practical calculations, the integral (2.1) must be simplified to a sum, as the solar photons flux is only known (or tabulated) for a discrete set of wavelengths intervals

$$J = \sum_i J_i = \sum_i \sigma_{\mathrm{max}_i}(\lambda_i)\Phi_i(\lambda_i)\Delta\lambda_i, \tag{2.2}$$

where $J_i$ is the photolysis rate, $\sigma_{\mathrm{max}i}(\lambda_i)$ is the photolysis cross-section at wavelength $\lambda_i$, $\Phi_i(\lambda_i)$ is the solar photon flux and $\Delta\lambda_i$ is the half bandwidth for the $i$th wavelength band. Experimental values for $\Phi_i(\lambda_i)$ can be taken, for example, from the works of Chance & Kurucz [38]. To estimate the solar flux in the troposphere, we set $\Phi_i(\lambda_i)$ to zero for $\lambda < 305$ nm.

The photolysis cross-section as a function of photon energy $\sigma(E)$ can be calculated as [30–34]

$$\sigma(E) = E \cdot |\ll \Psi_f(\mathbf{r},\mathbf{R})|\mathbf{d}|\Psi_i(\mathbf{r},\mathbf{R}) \gg|^2, \tag{2.3}$$

where $\Psi_f(\mathbf{r},\mathbf{R})$ is the vibronic wave function of the molecule in the final excited state, $\Psi_i(\mathbf{r},\mathbf{R})$ is the vibronic wave function of the molecule in the initial state, $E$ is the energy difference between the initial and final states and $\mathbf{d}$ is the dipole moment operator of the molecule, $\mathbf{r}$ represents all electronic coordinates and $\mathbf{R}$ all nuclear coordinates. For photolysis reactions, the relevant final excited states are usually unbound (i.e. the energy spectrum of the final states is continuous, as opposed to discrete in the initial bound state). Cross-sections can then be calculated in the framework of the adiabatic and Franck–Condon approximations. For diatomic molecules, assuming that the vibrational wave function of the initial state can be treated using the harmonic oscillator approximation, we obtain [33–35,39]

$$\sigma(E_{\mathrm{if}} + \varepsilon) = \frac{4\pi^2}{3\cdot c}(E_{\mathrm{if}} + \varepsilon)\cdot|(d_e^{\mathrm{if}})^2\cdot < \chi_{\mathrm{cont}}(R)|\chi_{\mathrm{in}}(R) > |^2 \tag{2.4}$$

$$< \chi_{\mathrm{cont}}(R)|\chi_{\mathrm{in}}(R) >=$$
$$= \int_0^\infty \frac{2}{\pi}\sqrt{\frac{m}{B}\cdot\sinh(\frac{2\pi\sqrt{2mE_k}}{B})}\cdot K_{i\cdot\frac{2\sqrt{2mE_k}}{B}}\left(\frac{2}{B}\sqrt{2mA}\cdot\exp\left(-\frac{B\cdot R}{2}\right)\right)\cdot\left(\frac{m\omega}{\pi}\right)^{1/4}\cdot\exp\left(-\frac{m\omega}{2}(R - R_0)^2\right)\mathrm{d}R.$$
$$\tag{2.5}$$

Here, $K$ is the Macdonald function and $\varepsilon$ is the energy which is counted from the predissociation limit. Expressions (2.4) and (2.5) are obtained by representing the potential energy curve of the unbound excited state as an analytical function $U(R) = A\cdot\exp(-BR)$, where $A$ and $B$ are constant parameters [33–35,40]. $m$ is the reduced mass of the molecule, $\omega$ is the harmonic frequency of molecular vibration in the ground electronic state, $R_0$ is the equilibrium position (or equilibrium bond length) of the molecule, $E_{\mathrm{if}}$ is the energy of dissociation to the electronic state $f$ (i.e. the energy difference between the reactant in state $i$ and the infinitely separated photolysis products in state $f$) and $d_e^{\mathrm{if}}$ is the electric dipole moment of the vertical electronic transition between pure spin electronic states. Expressions (2.4) and (2.5) can be extended to polyatomic species as discussed below.

As an illustrative example of how to apply expressions (2.4) and (2.5) to the polyatomic molecules studied here, we consider first the photolysis of $CH_3O\dot{O}$, i.e. the photochemical reaction $CH_3O\dot{O} + h\nu \rightarrow CH_3\dot{O} + O(^3P)$. Note that the formation of triplet oxygen is spin-allowed, as a doublet and a triplet can couple to a doublet. A preliminary extended quasi-degenerate second-order multireference perturbation theory (XMC-QDPT2) *ab initio* calculation (see below for calculation details) shows that the first excited state of $CH_3O_2$ is bound, but the next four states are unbound,

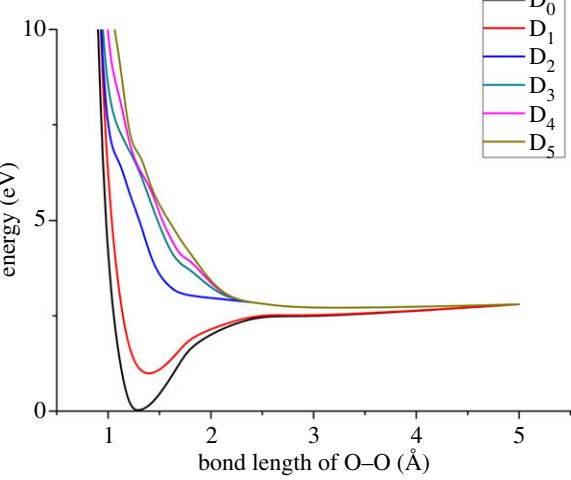

**Figure 1.** The potential energy curves corresponding to the first predissociation limit for $CH_3O_2 + h\nu => CH_3O + O$, computed using XMC-QDPT2(11,7)/cc-pv5z. Note: only the O–O bond is kept fixed, and all other coordinates are relaxed in the ground state. The excited-state energies are then calculated at the ground state geometry.

and correspond to the first predissociation limit and O–O bond breaking. The potential energy curves of the ground and first five excited states (as a function of the O–O distance) are shown in figure 1. The other excited states are located significantly higher (more than 8 eV) and correspond to the formation of $O(^1D)$. The only electronic transition with a large electronic dipole transition moment is $D_0 \rightarrow D_2$, with a transition dipole moment of 0.67 atomic units (a.u.). The other transitions have transition dipole moments of less than 0.001 a.u. Thus, we only need to consider photodissociation into the unbound state via the $D_0 \rightarrow D_2$ electronic transition. $\omega$ can be calculated using the second derivative of the $D_0$ potential energy curve shown in figure 1, while $m$ can be estimated using the expression $m = (m_{CH_3O} \cdot m_O)/(m_{CH_3O} + m_O)$. (In practice, we are thus treating the polyatomic molecule as 'effectively diatomic', with the two fragments connected by the bond broken in the photolysis reaction.) Thus, all parameters required for the evaluation of expressions (2.4) and (2.5) can be obtained using quantum chemical methods. The numerical values for the parameters $A$ and $B$ are obtained by fitting to the computed potential energy curve of state $D_2$, as $D_0 \rightarrow D_2$ has the largest electronic transition dipole moment.

## 2.2. The photophysics and chemistry in the $D_1$ state

In order to check the hypothesis about $RO_2$ photolysis by IR radiation, we calculated all the deactivated energy channels of the $D_1$ excited state. These are the radiative ($k_r$) and radiationless IC ($k_{IC}$) processes. $k_r$ is calculated using the Strickler–Berg equation [41]

$$k_r = \frac{1}{1.5} \cdot f \cdot E^2(D_1 \rightarrow D_0), \tag{2.6}$$

where $f$ is the oscillator strength and $E(D_1 \rightarrow D_0)$ is the de-excitation energy from $D_1$ to $D_0$. Note that in the original formula [41], the integral over the absorption band is used, but here it is assumed that the electronic transition energy can be directly used instead.

$k_{IC}$ is calculated using the algorithm described in [40,42] with the formula

$$k_{IC} = \frac{4}{\Gamma_f} \sum_{n_1,n_2,\ldots,n_{3N-6}}^{E_{if}=n_1\omega_1+n_2\omega_2+\ldots n_{3N-6}\omega_{3N-6}} \left( D \prod_{k=1}^{3N-6} \left( \frac{e^{-y_k}y_k^{n_k}}{n_k!} \right)^{1/2} + \left[ \sum_{j=1}^{3N-6} d_j \cdot \mathrm{non}_j \prod_{\substack{k=1\\k\neq j}}^{3N-6} \left( \frac{e^{-y_k}y_k^{n_k}}{n_k!} \right)^{1/2} \right] + \sum_j \sum_{j'} \mathrm{non}_j h t_{j'} W_{jj'} \cdot \prod_{\substack{k=1\\k\neq j\\k\neq j'}}^{3N-6} \left( \frac{e^{-y_k}y_k^{n_k}}{n_k!} \right)^{1/2} \right)^2 \tag{2.7}$$

where

$$W_{jj'} = -\sum_{v}\sum_{q}\sum_{v'}\sum_{q'} < \frac{\phi_i(\boldsymbol{r,s,R})\partial^2\phi_f(\boldsymbol{r,s,R})}{\partial R_{vq}R_{v'q'}} > |_{\boldsymbol{R}=\boldsymbol{R}_0} M_v^{-1/2}M_{v'}^{-1/2}L_{vqj}L_{v'q'j'} \tag{2.8}$$

$$D = -\sum_{v}\sum_{q}(2M_v)^{-1} < \phi_i(\boldsymbol{r,s,R})|\frac{\partial^2}{\partial R_{vq}^2}|\phi_f(\boldsymbol{r,s,R}) > |_{\boldsymbol{R}=\boldsymbol{R}_0}, \tag{2.9}$$

$$d_j = -\sum_{v}\sum_{q}M_v^{-1/2}L_{vqj} < \phi_i(\boldsymbol{r,s,R})\frac{\partial\phi_f(\boldsymbol{r,s,R})}{\partial R_{vq}} > |_{\boldsymbol{R}=\boldsymbol{R}_0}, \tag{2.10}$$

$$ht_j = < \chi_{i0_j}(Q_j)|Q_j|\chi_{fn_j}(Q_j) > = \left[\frac{1}{2\omega_j n_j!}(n_j + y_j)^2 \cdot \mathrm{e}^{-y_j} \cdot y_j^{n_j-1}\right]^{1/2}, \tag{2.11}$$

$$\mathrm{non}_j = < \chi_{i0_j}(Q_j)|\frac{\partial}{\partial Q_j}|\chi_{fn_j}(Q_j) > = \left[\frac{1}{2n_j!}\omega_j(n_j - y_j)^2 \cdot \mathrm{e}^{-y_j} \cdot y_j^{n_j-1}\right]^{1/2}, \tag{2.12}$$

$$y_j = \frac{1}{2}(\omega_j) \cdot |Q_{0_j}^f - Q_{0_j}^i|^2, \tag{2.13}$$

$$W_j = \left[\sum_{v}\sum_{q}\frac{\partial(H_{SO}^{if})}{\partial R_{vq}}|_{\boldsymbol{R}=\boldsymbol{R}_0}M_v^{-1/2}L_{vqj}\right]. \tag{2.14}$$

Here, $< \frac{\varphi_i(\boldsymbol{r,s,R})\partial^2\varphi_f(\boldsymbol{r,s,R})}{\partial R_{vq}R_{v'q'}} > |_{\boldsymbol{R}=\boldsymbol{R}_0}$ and $< \varphi_i(\boldsymbol{r,s,R})\frac{\partial\varphi_f(\boldsymbol{r,s,R})}{\partial R_{vq}} > |_{\boldsymbol{R}=\boldsymbol{R}_0}$ are the non-adiabatic coupling matrix elements (NACME) of second order and first order, respectively. $M_v$ is the mass of the $v$th atom, and $L_{vqj}$ are coefficients of the linear relation between the Cartesian ($R$) and the normal coordinates ($Q$): $R_{vq} - R_0 vq = M_v^{-1/2}L_{vqj}Q_j$. The $ht_j$ and $\mathrm{non}_j$ are Herzberg–Teller and non-adiabatic factors. $y_j$ is the Huang–Rhys factor of the $j$th promotive mode. $E_{if}$ is the energy gap between initial and final states, and $n_j$ and $\omega_j$ are the excitation number and the frequency of the $j$th mode, respectively.

Finally, the rate constant of the decay of the $D_1$ state is estimated as

$$k_{D_1} = k_{IC} + k_r. \tag{2.15}$$

H-shift reactions for $CH_3O\dot{O}$ and $COHCH_2O\dot{O}$ were also studied in the $D_0$ and $D_1$ states. In the former case, we studied the 1,3 H-shift leading to the formation of $H\dot{O}$ and $CH_2\dot{O}$. In the latter case, we studied the aldehydic 1,4 H-shift, which leads to the formation of a $COCH_2OOH$ alkyl radical, which may subsequently either decompose to $CO + CH_2OOH$ (with the latter further decomposing to $CH_2O + \dot{O}H$) or add an $O_2$ molecule to give $OOCOCH_2OOH$. This H-shift has a barrier of roughly 25 kcal mol$^{-1}$ in the ground state, and is intended to be representative of the potentially even faster H-shifts of larger $RO\dot{O}$ involved in autoxidation [11].

The rate constant of this reaction was estimated using elementary transition state theory [43]

$$k = \kappa\frac{k_B T}{h}\frac{Q_{TS}}{Q_R}\exp\left(-\frac{E_b}{k_B T}\right), \tag{2.16}$$

where $Q_{TS}$ and $Q_R$ are the partition functions of transition and reactant structures, respectively. $k_B$ is Boltzmann's constant, $h$ is Planck's constant, $\kappa$ is the Eckart correction factor for tunnelling, $T$ is temperature and $E_b$ is threshold energy. The $\kappa$ is estimated using the scheme described in [44], where the activation barriers of forward and back reactions, the reduced mass and imaginary frequencies are required. We note that activation barrier for the back reaction was obtained to be approximately 70 kcal mol$^{-1}$ for $CH_3O\dot{O}$. This value was used for the estimation of $\kappa$ using (2.16).

## 2.3. Calculation details

### 2.3.1. Photolysis calculations

We consider the following five peroxyl radicals: methyl peroxyl radical ($CH_3O\dot{O}$), ethyl peroxyl radical ($CH_3CH_2O\dot{O}$), 2-oxoethyl peroxyl radical ($C(O)HCH_2OO$), methyl acylperoxyl radical ($HC(O)O\dot{O}$) and ethyl acylperoxyl radical ($CH_3C(O)O\dot{O}$). They are shown in figure 2. These are important atmospheric species in their own right, and also provide a representative sample of the functional groups typically found in more complex peroxyl radicals: alkyl and carbonyl groups. As carbonyl groups are known to act as chromophores, they could be speculated to have a significant effect on the photolysis cross-sections. Similarly, acyl peroxyl radicals often exhibit very different chemical reactivity from the

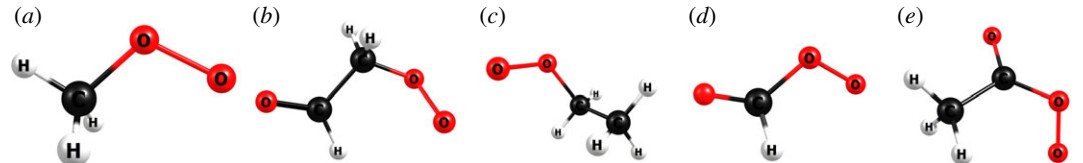

**Figure 2.** The peroxyl radicals: (*a*) CH$_3$O$_2$; (*b*) COHCH$_2$O$_2$; (*c*) CH$_3$CH$_2$O$_2$; (*d*) COHO$_2$; and (*e*) CH$_3$COO$_2$. Colour coding: O, red; C, black; H, white.

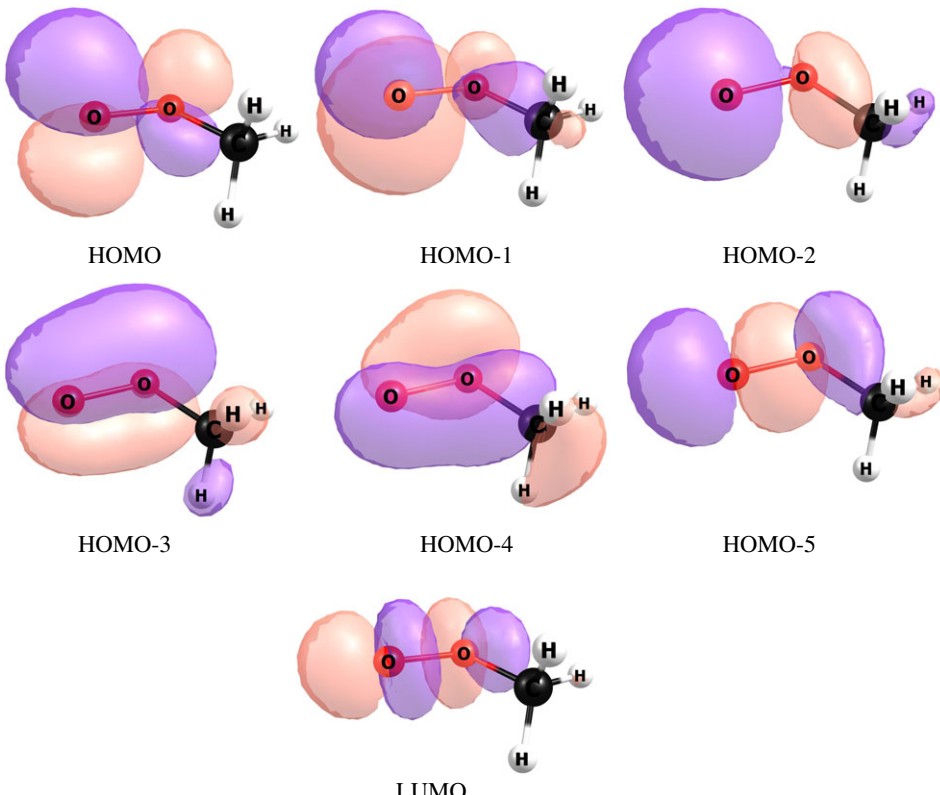

HOMO

HOMO-1

HOMO-2

HOMO-3

HOMO-4

HOMO-5

LUMO

**Figure 3.** The MOs of the active space used for CH$_3$O$_2$.

corresponding 'regular' peroxyl radicals [6]—it is interesting to see if this difference extends also to photochemistry.

The complete active space self-consistent field (CASSCF) included 11 electrons in seven molecular orbitals (MOs), and the state averaging (SA) was performed for the six lowest electronic states, which are well separated from higher states for all the considered radicals; CH$_3$OȮ, COHCH$_2$OȮ, CH$_3$CH$_2$OȮ, COHOȮ and CH$_3$COOȮ. These active spaces were quite stable with respect to variations in the O–O nuclear distance for the peroxyl radicals. The active space for CH$_3$OȮ is shown in figure 3. HOMO and HOMO-1 correspond to O atom lone pairs, HOMO-2 is a $\sigma$-bonding MO for the O–C bond, HOMO-3 and HOMO-4 are O–O $\pi$-bonding MOs, and HOMO-5 and LUMO are $\sigma$-bonding and $\sigma^*$-antibonding MOs of the O–O bond. We also tested including the $\sigma^*$-antibonding MO of the O–C bond in the active space (following the general rule-of-thumb that bonding–antibonding pairs should both be included), but this led to instability of the active space as the O–O distance was varied. The orbital in question was, therefore, omitted. The active space used for the other radicals consists of the same MOs as in figure 3.

The cc-pv5z basis set was used for all calculations [45]. The potential energies curves of the electronic states were plotted at the XMC-QDPT2 [46] level with the relaxation surface option and a step size of 0.01 Å step along O–O bond for all molecules. In other words, the O–O bond length in the peroxyl radicals was scanned along the ground electronic state, relaxing all nuclear other coordinates, and vertical transitions to the considered excited electronic states were then computed for each point on the scan.

**Table 1.** The photolysis data of $CH_3O_2$, $COHCH_2O_2$, $CH_3CH_2O_2$, $COHO_2$ and $CH_3COO_2$ molecules, computed at the XMC-QDPT2(11,7)/cc-pv5z level.

| radical | transition dipole moment, a.u. | $J_{troposhere}$ |
|---|---|---|
| $CH_3O\dot{O}$ | 0.66 $(D_0 \rightarrow D_2)$ | $1.6 \times 10^{-5}$ s$^{-1}$ |
| $COHCH_2O\dot{O}$ | 1.00 $(D_0 \rightarrow D_2)$ | $3 \times 10^{-5}$ s$^{-1}$ |
| $CH_3CH_2O\dot{O}$ | 0.67 $(D_0 \rightarrow D_2)$ | $1.3 \times 10^{-5}$ s$^{-1}$ |
| $COHO\dot{O}$ | 0.59 $(D_0 \rightarrow D_2)$ | $1.1 \times 10^{-5}$ s$^{-1}$ |
| $CH_3COO\dot{O}$ | 0.64 $(D_0 \rightarrow D_2)$ | $1.5 \times 10^{-5}$ s$^{-1}$ |

Nuclear coordinates in the excited states were not relaxed. The effective Hamiltonian included 35 states in the XMC-QDPT2 calculation. All calculations were carried out using the FIREFLY software [47]. The data of photolysis cross-section calculation are given in electronic supplementary material, S1 chapter. It should be noted that using smaller basis sets such as 6–31++G(d,p) or even 6–311++G(d,p) led to large discrepancies in the peak cross-photolysis sections compared to the experimental one. For example, using the 6–311++G(d,p) basis set (with all other computational details as described above) led to the absence of any notable cross-sections in the photolysis spectrum beyond 305 nm, which is extremely important for the tropospheric photolysis lifetime estimation. The reason for this is the insufficient accuracy of the potential energy surface at long nuclear distances. To correctly treat this system, a basis set larger than cc-pvqz is required.

### 2.3.2. Photophysics calculation

The NACME between $D_0$ and $D_1$ was calculated at the time-dependent density functional theory (TDDFT) level of theory using the response formalism [48]. TDDFT is applicable to this process, as DFT functionals account for dynamic correlation, and our XMC-QDPT2 results indicate that the optimized $D_1$ state is not a multireference system (i.e. static correlation is not important). We chose the B3LYP functional, as previous studies indicate that it gives correct NACME values for excitation energies of single molecules, provided that the excitations do not involve charge transfer [40,42].

The hessian in the $D_1$ state and gradient in the $D_0$ state were calculated using the same method. The oscillator strength ($f$) was calculated at the same level of theory. The basis set was the same as previously. These calculations were performed in Turbomole software [49].

### 2.3.3. H-shift rate calculations

The transition states and products of the H-shift reactions were calculated at the XMC-QDPT2 level using the 6–311++G(d,p) basis set, in both $D_0$ and $D_1$ states. The H-shift calculation was also carried out using the cc-vptz basis set for $CH_3O\dot{O}$ in both $D_0$ and $D_1$ states in order to validate the choice of basis set. These calculations were performed in Firefly.

# 3. Results and discussion

## 3.1. Photolysis of peroxyl radicals in ultraviolet region

As in our previous calculations [33–35], the variations observed between the photolysis cross-sections of the five studied peroxyl radicals mainly arise from differences in the transition dipole moments. The cross-sections are thus roughly proportional to the squares of the transition dipole moments.

Table 1 shows the transition dipole moments and $J$ values of the five studied peroxyl radicals. The wavelength-dependent photolysis cross-section for $CH_3O\dot{O}$ is also shown in figure 4 (data for other peroxyl radicals are similar, and are given in electronic supplementary material, section S1). The transition dipole moments of the different peroxyl radicals vary by less than a factor of 2, and the photolysis rates by less than a factor of 3. This demonstrates that the photolysis process for peroxyl radicals is essentially local: adjacent functional groups have relatively little effect on the photolysis rate. It is very likely that also larger peroxyl radicals have transition dipole moments and photolysis rates (at least for the $RO\dot{O} => R\dot{O} + O(^3P)$ channel) of the same order of magnitude as those studied here. It should be noted that while the maximum of the photolysis cross-section is found around

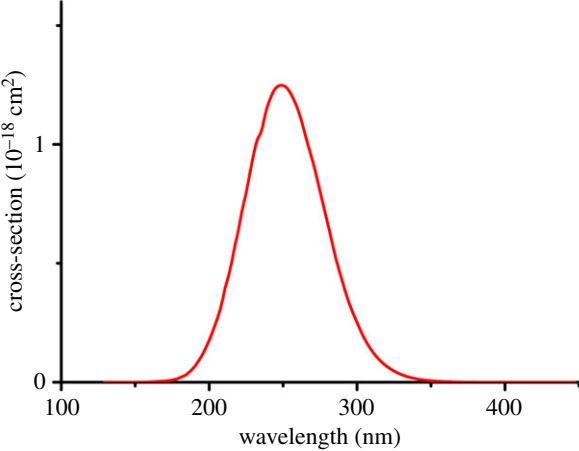

**Figure 4.** The cross-section of $CH_3O_2$ photolysis corresponding to the first dissociation limit at XMC-QDPT2(11,7)/cc-pv5z.

**Table 2.** IC rates ($k_{IC}$) and reaction rates (OH loss, $k_{OH}$, HO$_2$, $k_{HO2}$, or 1,4 H-shift, $k_{H-shift}$) of $CH_3O_2$, $COHCH_2O_2$, $CH_3CH_2O_2$, $COHO_2$ and $CH_3COO_2$ molecules in the D$_1$ excited state, computed at the XMC-QDPT2(11,7)/6–311++G(d,p) level.

| radical | $k_{IC}$ | $k_{reaction}$ |
|---|---|---|
| $CH_3O\dot{O}$ | $1.1 \times 10^{13}$ s$^{-1}$ | $k_{OH} = 1.0 \times 10^{-12}$ s$^{-1}$ |
| $COHCH_2O\dot{O}$ | $1.6 \times 10^{13}$ s$^{-1}$ | $k_{H-shift} = 5.0 \times 10^{-7}$ s$^{-1}$ |
| $CH_3CH_2O\dot{O}$ | $1.1 \times 10^{13}$ s$^{-1}$ | $k_{H-shift} = 4.2 \times 10^{-6}$ s$^{-1}$ |
| $COHO\dot{O}$ | $1.1 \times 10^{13}$ s$^{-1}$ | $k_{HO2} = 1.1 \times 10^{-18}$ s$^{-1}$ |
| $CH_3CO O\dot{O}$ | $3.0 \times 10^{13}$ s$^{-1}$ | $k_{H-shift} = 5.0 \times 10^{-8}$ s$^{-1}$ |

248 nm (with values around $1.2 \times 10^{-18}$ cm$^2$), part of the spectrum extends from 300 nm until about 349 nm. Our results agree reasonably with the experimental measurements of photolysis at least for $C_2H_5O_2$, for which many studies have found a peak located around 240 nm, with values of 3–5 $\times$ $10^{-18}$ cm$^2$ [15,17–28]. Thus, our model works well for the photolysis cross-section estimation for polyatomic molecules. Also, we note that when using the 6–311++G(d,p) basis set, the peaks were located at shorter wavelengths (around 204 nm), and as a consequence, the cross-section photolysis was (incorrectly) predicted to be essentially zero for wavelengths longer than 305 nm (see electronic supplementary material, figure S1).

In the troposphere, the photolysis lifetime ($1/J$) for the studied peroxyl radicals varies between 9.3 and 25 h. This is many orders of magnitude longer than the lifetime of peroxyl radicals with respect to their main sink reactions with HO$_2$, NO and other RO$_2$, which typically is of the order of 1–100 s [11].

## 3.2. Photolysis of peroxyl radicals in infrared region

The results of the H-shift and IC calculations are summarized in table 2 and electronic supplementary material table S2.1. The calculated $k_r$ values are around $10^2$–$10^3$ s$^{-1}$. Radiative decay is thus fully negligible, and the IC process dominates the D$_1$ decay. The lifetime of the D$_1$ state with respect to IC is around $10^{-13}$ s for all studied RO$\dot{O}$. This is comparable with typical timescales for vibrational relaxation, or with the frequency of molecular vibrations ($10^{-14}$ s). For the excited (D$_1$) states of $CH_3O\dot{O}$ and $COHO\dot{O}$, we could not find a transition state (TS) for 1,3 H-shifts, but found instead a TS for the subsequent OH and HO$_2$ formation, respectively. We, therefore, used these TS to estimate an upper limit for the rate of the overall $CH_3O\dot{O}$ => $CH_2O + \dot{O}H$ (and $COHO\dot{O}$ => $CO + HO\dot{O}$) process in the excited state. Should the actual H-shifts have even higher barriers, the real rates would correspondingly be lower. For the three other RO$_2$, we found transition states for the 1,4 H-shifts. All activation barriers and rate constants ($k_{OH}$, $k_{HO2}$ and $k_{H-shift}$) are given in the electronic supplementary material, as are Cartesian coordinates and illustrations of the reactants and transition states. The computed rate constants ($k_{OH}$ and $k_{H-shift}$) are generally higher in the D$_1$ state than in the D$_0$ state, as

expected. The exception to this rule is $COHCH_2O_2$, where the rate (as usual for aldehydic H-shifts) is fast in the $D_0$ state with $k_{H\text{-}shift} = 4.8$ s$^{-1}$, but only $5.0 \times 10^{-7}$ s$^{-1}$ in the $D_1$ state. Thus, our initial hypothesis, based on the proposal of Frost [12], that fast H-shifts could be even faster in the excited state, seems to be incorrect. Our results indicate that the excitation does not necessarily lower reaction barriers, but may even raise them, as shown for $COHCH_2O\dot{O}$. Unexpectedly, the fastest reaction rate found in the $D_1$ state is $k_{H\text{-}shift} = 4.2 \times 10^{-6}$ s$^{-1}$ for $CH_3CH_2O\dot{O}$. However, even these enhanced H-shift rates are negligible compared to the IC rate with $k_{IC} \sim 10^{13}$ s$^{-1}$. Thus, the lifetime of the $D_1$ state is too short to allow for H-shifts and subsequent dissociation (or alkyl radical formation, $O_2$ addition and autoxidation). Also, we note that using cc-vptz basis set, which is somewhat larger than 6–311++G(d, p), gives almost the same result for both $k_{H\text{-}shift}$ and $k_{OH}$ for $CH_3O\dot{O}$ (see electronic supplementary material, table S2.2).

## 4. Conclusion

We have extended our recently published approach for computing photolysis rates of diatomic molecules to five different atmospherically relevant peroxyl radicals. Due to the higher energy required for the $RO\dot{O} + h\nu => R\dot{O} + O(^3P)$ reactions, the photolysis rates of all studied peroxyl radicals in the troposphere are very low; less than $10^{-5}$ s$^{-1}$. Photolysis can thus be safely and completely ignored as a loss term for peroxyl radicals in the troposphere. We also investigated whether IR excitation to the lowest excited $D_1$ state can promote H-shift reactions and thus affect $RO\dot{O}$ reactivity in the atmosphere. While excitation indeed lowers the barrier for some H-shift reactions, the lifetime of the $D_1$ state with respect to IC was found to be only $10^{-13}$ s for all studied $RO\dot{O}$. Even the enhanced H-shifts are thus unable to compete with decay back to the ground state, and IR excitation is thus unlikely to affect atmospheric $RO\dot{O}$ reactivity.

Data accessibility. The details of our calculations were included in the electronic supplementary material. Therefore, any user can reproduce our results using this file and the text of our manuscript.

Authors' contributions. The calculations were carried out by R.R.V. The writing of manuscript and the research purpose of the present project were formed by T.K. and R.R.V.

Competing interests. We declare we have no competing interests.

Funding. We received no funding for this study.

Acknowledgements. We thank the Academy of Finland—1325369 and 1315600, and the CSC IT Center for Science in Espoo, Finland, for computing time.

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
