## [Reviewer comments · Royal Society Open Science]

Review History

RSOS-200521.R0 (Original submission)

Review form: Reviewer 1

Is the manuscript scientifically sound in its present form?

Yes

Are the interpretations and conclusions justified by the results?

Yes

Is the language acceptable?

Yes

Do you have any ethical concerns with this paper?

No

Have you any concerns about statistical analyses in this paper?

No

Recommendation?

Accept with minor revision (please list in comments)

Comments to the Author(s)

This is a logical study that addresses a potentially important process for atmospheric chemistry. The manuscript is generally well written but the conclusions are rather disappointing. Nonetheless, it is important to consider the possible fate of peroxy radicals and the authors have provided a convincing account of the likely photochemistry. Minor points to consider are: (i) Peroxide radicals in the title should read peroxy radicals and (ii) the Strickler-Berg expression was not developed for such weakly absorbing species. This might not be a valid application. Finally, peroxy radicals are good oxidants and are able to abstract hydrogen atoms from many substrates. Are such reactions important here as competing processes?

Review form: Reviewer 2

Is the manuscript scientifically sound in its present form?

Yes

Are the interpretations and conclusions justified by the results?

Yes

Is the language acceptable?

Yes

Do you have any ethical concerns with this paper?

No

Have you any concerns about statistical analyses in this paper?

No

Recommendation?

Accept with minor revision (please list in comments)

Comments to the Author(s)

This is a nice and interesting study of the fate of peroxy radicals in the troposphere, carried out using highly correlated post-HF methods. Particularly, the authors investigate the photolysis channel and come to the conclusion that it is not a competitive process. Subsequently, they study H shift reactions in the D1 excited state, but also in this case they come to the conclusions that they are unlikely to occur due to the fast decay to D0. The paper is well written and the scientific contents is sound. I have no reason to suspect about the correctness of the results, which are rigorously described and all computational strategies are fully and properly justified.

Few minor points:

- I would use the symbol of radicals in the formulas in the text. Although it is boring to add, it is chemically more correct
- I would introduce the H shift investigation in the title too, since it is an important aspect of the work.

Decision letter (RSOS-200521.R0)

Dear Dr Valiev:

Title: Is Photolysis of Peroxide Radicals a Competitive Pathway in the Troposphere?
Manuscript ID: RSOS-200521

Thank you for submitting the above manuscript to Royal Society Open Science. On behalf of the Editors and the Royal Society of Chemistry, I am pleased to inform you that your manuscript will be accepted for publication in Royal Society Open Science subject to minor revision in accordance with the referee suggestions. Please find the reviewers' comments at the end of this email.

The reviewers and handling editors have recommended publication, but also suggest some minor revisions to your manuscript. Therefore, I invite you to respond to the comments and revise your manuscript.

Because the schedule for publication is very tight, it is a condition of publication that you submit the revised version of your manuscript before 30-May-2020. Please note that the revision deadline will expire at 00.00am on this date. If you do not think you will be able to meet this date please let me know immediately.

Kind regards,
Dr Laura Smith
Publishing Editor, Journals

On behalf of the Subject Editor Professor Anthony Stace and the Associate Editor Dr Nadia Martinez Villegas.

RSC Associate Editor:

Comments to the Author:

The research presented in this draft is original and might be of interest to RSOS audience. Although conclusions are supported by experimental data, the length of the manuscript should be shortened. Some of the material (about the various peroxide radicals) should be presented as an electronic appendix, while the use of radical symbols in the formulas in the text is kindly requested. Additionally, the title should be modified according to the Reviewers suggestions.

RSC Subject Editor:

Comments to the Author:

(There are no comments.)

Reviewer comments to Author:

Reviewer: 1

Comments to the Author(s)

This is a logical study that addresses a potentially important process for atmospheric chemistry. The manuscript is generally well written but the conclusions are rather disappointing. Nonetheless, it is important to consider the possible fate of peroxy radicals and the authors have provided a convincing account of the likely photochemistry. Minor points to consider are: (i) Peroxide radicals in the title should read peroxy radicals and (ii) the Strickler-Berg expression was not developed for such weakly absorbing species. This might not be a valid application. Finally, peroxy radicals are good oxidants and are able to abstract hydrogen atoms from many substrates. Are such reactions important here as competing processes?

Reviewer: 2

Comments to the Author(s)

This is a nice and interesting study of the fate of peroxy radicals in the troposphere, carried out using highly correlated post-HF methods. Particularly, the authors investigate the photolysis channel and come to the conclusion that it is not a competitive process. Subsequently, they study H shift reactions in the D1 excited state, but also in this case they come to the conclusions that they are unlikely to occur due to the fast decay to D0. The paper is well written and the scientific

contents is sound. I have no reason to suspect about the correctness of the results, which are rigorously described and all computational strategies are fully and properly justified.

Few minor points:

- I would use the symbol of radicals in the formulas in the text. Although it is boring to add, it is chemically more correct
- I would introduce the H shift investigation in the title too, since it is an important aspect of the work.

Author's Response to Decision Letter for (RSOS-200521.R0)

See Appendix A.

RSOS-200521.R1 (Revision)

Review form: Reviewer 1

Is the manuscript scientifically sound in its present form?

Yes

Are the interpretations and conclusions justified by the results?

Yes

Is the language acceptable?

Yes

Do you have any ethical concerns with this paper?

No

Have you any concerns about statistical analyses in this paper?

No

Recommendation?

Accept as is

Comments to the Author(s)

The revised manuscript addresses the concerns raised earlier and is now suitable for publication.

Review form: Reviewer 2

Is the manuscript scientifically sound in its present form?

Yes

Are the interpretations and conclusions justified by the results?

Yes

Is the language acceptable?

Yes

Do you have any ethical concerns with this paper?

No

Have you any concerns about statistical analyses in this paper?

No

Recommendation?

Accept with minor revision (please list in comments)

Comments to the Author(s)

The revision of the manuscript is satisfactory. Please notice that there is a mistake in the title, which should read '... A Pathway....' and not '..A Pathways...'

Decision letter (RSOS-200521.R1)

Dear Dr Valiev:

Title: Is Either Direct Photolysis or Photocatalyzed H-shift of Peroxyl Radicals A Competitive Pathways in the Troposphere?

Manuscript ID: RSOS-200521.R1

It is a pleasure to accept your manuscript in its current form for publication in Royal Society Open Science. The chemistry content of Royal Society Open Science is published in collaboration with the Royal Society of Chemistry.

On behalf of the Subject Editor Professor Anthony Stace and the Associate Editor Dr Nadia Martinez Villegas.

RSC Associate Editor:
Comments to the Author:

Please be aware that there is a mistake in the title, which should be corrected during proofreading. I should read '... A Pathway....' and not '..A Pathways...'

RSC Subject Editor:

Comments to the Author:

(There are no comments.)

Reviewer(s)' Comments to Author:

Reviewer: 1

Comments to the Author(s)

The revised manuscript addresses the concerns raised earlier and is now suitable for publication.

Reviewer: 2

Comments to the Author(s)

The revision of the manuscript is satisfactory. Please notice that there is a mistake in the title, which should read '... A Pathway....' and not '..A Pathways...'

Appendix A

We thank both reviewers for their work, which has helped improve the manuscript. Reviewer comments are reproduced below in black font, with our reviews in red bolded font.

Reviewer: 1

Comments to the Author(s)

This is a logical study that addresses a potentially important process for atmospheric chemistry. The manuscript is generally well written but the conclusions are rather disappointing. Nonetheless, it is important to consider the possible fate of peroxy radicals and the authors have provided a convincing account of the likely photochemistry. Minor points to consider are:

(i) Peroxide radicals in the title should read peroxy radicals and

Reply:

We changed the notation and now use the notation “peroxy” for the studied radicals.

(ii) the Strickler-Berg expression was not developed for such weakly absorbing species. This might not be a valid application.

Reply:

We agree with the referee that the the Strickler-Berg expression works well for strong absorption. However, this expression has been applied many times also for weak absorption [<https://doi.org/10.1016/j.cplett.2019.136914>, <https://doi.org/10.1039/C6CP03060B>, <https://doi.org/10.1039/C9CP03183A>, <https://doi.org/10.1039/C7CP08703A>], even for a case where the transition is forbidden (the case of free base porphyrin where oscillator strength (f) was taken from the experimental work $f=0.03$ [<https://doi.org/10.1039/C9CP03183A>]). The expression can predict the true order of magnitude for radiative rate constant (kr) even in these cases. In any case, our result show that the radiative channel is completely negligible for peroxy radicals in comparison with the internal conversion process.

Finally, peroxy radicals are good oxidants and are able to abstract hydrogen atoms from many substrates. Are such reactions important here as competing processes?

Reply:

Yes, unimolecular H-abstractions are competing processes for complex enough peroxyradicals (e.g. with aldehyde groups). This is mentioned in the manuscript, and indeed one of the reaction types we study is a (photocatalyzed) H-abstraction. However, BIMOLECULAR peroxy H-abstractions, while important in the condensed phase, are not competitive in the gas phase - the reactions inevitably have barriers of at least 10 kcal/mol or so (usually more like 20 kcal/mol), which means that only a very very small

fraction of collisions leads to reaction - and thus something else will happen to the radicals instead (in practice, usually an almost or completely barrierless bimolecular reaction such as reaction with NO, HO₂, or other RO₂).

Referee: 2

Comments to the Author

This is a nice and interesting study of the fate of peroxy radicals in the troposphere, carried out using highly correlated post-HF methods. Particularly, the authors investigate the photolysis channel and come to the conclusion that it is not a competitive process. Subsequently, they study H shift reactions in the D1 excited state, but also in this case they come to the conclusions that they are unlikely to occur due to the fast decay to D0. The paper is well written and the scientific contents is sound. I have no reason to suspect about the correctness of the results, which are rigorously described and all computational strategies are fully and properly justified.

Few minor points:

- I would use the symbol of radicals in the formulas in the text. Although it is boring to add, it is chemically more correct

Reply:

we introduced this symbol in the main text of manuscript.

- I would introduce the H shift investigation in the title too, since it is an important aspect of the work.

Reply:

We changed the title. New title is “Is Either Direct Photolysis or Photocatalyzed H-shift of Peroxide Radicals A Competitive Pathways in the Troposphere?”